# Dual Role of NMDAR Containing NR2A and NR2B Subunits in Alzheimer’s Disease

**DOI:** 10.3390/ijms25094757

**Published:** 2024-04-26

**Authors:** Iu Raïch, Jaume Lillo, Joan Biel Rebassa, Toni Capó, Arnau Cordomí, Irene Reyes-Resina, Mercè Pallàs, Gemma Navarro

**Affiliations:** 1Centro de Investigación Biomédica en Red de Enfermedades Neurodegenerativas (CiberNed), National Institute of Health Carlos III, 28029 Madrid, Spain; iuraichipanisello@gmail.com (I.R.); lillojaume@gmail.com (J.L.); joanbrebassa@gmail.com (J.B.R.); ireyesre8@ub.edu (I.R.-R.); 2Institut de Neurociències UB, Campus Mundet, Passeig de la Vall d’Hebron 171, 08035 Barcelona, Spain; pallas@ub.edu; 3Department of Biochemistry and Physiology, Faculty of Pharmacy and Food Science, University of Barcelona, 08028 Barcelona, Spain; antoniocapoquetglas.98@gmail.com; 4Department of Biochemistry and Molecular Biomedicine, Faculty of Biology, University of Barcelona, 08028 Barcelona, Spain; 5Bioinformatics, Escola Superior de Comerç Internacional-University Pompeu Fabra (ESCI-UPF), 08003 Barcelona, Spain; arnau.cordomi@gmail.com; 6Pharmacology Section, Department of Pharmacology, Toxicology, and Therapeutic Chemistry, Faculty of Pharmacy and Food Sciences, Av Joan XXIII 27-31, 08028 Barcelona, Spain

**Keywords:** NMDA, NR2A, NR2B, Alzheimer’s disease, NMDAR, Aβ oligomers, pTau

## Abstract

Alzheimer’s disease (AD) is the main cause of dementia worldwide. Given that learning and memory are impaired in this pathology, NMDA receptors (NMDARs) appear as key players in the onset and progression of the disease. NMDARs are glutamate receptors, mainly located at the post-synapse, which regulate voltage-dependent influx of calcium into the neurons. They are heterotetramers, and there are different subunits that can be part of the receptors, which are usually composed of two obligatory GluN1 subunits plus either two NR2A or two NR2B subunits. NR2A are mostly located at the synapse, and their activation is involved in the expression of pro-survival genes. Conversely, NR2B are mainly extrasynaptic, and their activation has been related to cell death and neurodegeneration. Thus, activation of NR2A and/or inactivation of NR2B-containing NMDARS has been proposed as a therapeutic strategy to treat AD. Here, we wanted to investigate the main differences between both subunits signalling in neuronal primary cultures of the cortex and hippocampus. It has been observed that Aβ induces a significant increase in calcium release and also in MAPK phosphorylation signalling in NR2B-containing NMDAR in cortical and hippocampal neurons. However, while NR2A-containing NMDAR decreases neuronal death and favours cell viability after Aβ treatment, NR2B-containing NMDAR shows higher levels of cytotoxicity and low levels of neuronal survival. Finally, it has been detected that NMDAR has no effect on pTau axonal transport. The present results demonstrate a different role between GluNA and GluNB subunits in neurodegenerative diseases such as Alzheimer’s.

## 1. Introduction

Alzheimer’s disease (AD) is a neurodegenerative disorder that affects memory and other cognitive functions [1] and has become one of the most burdening diseases in this century [2]. The disease is classified into familial AD and sporadic AD [3]. Familial AD is induced by mutations in β-amyloid precursor protein and presenilin 1/2, and accounts only for 0.5% of AD cases, while the sporadic form is caused by hereditary together with environmental risk factors and is the dominant type of AD [4].

The pathophysiology of AD implies the presence of abnormal protein deposition in the brain in extracellular plaques of β-amyloid (Aβ) and intracellular neurofibrillary tangles (NFTs) [5]. Aβ monomers are derived from the cleavage of the neural membrane amyloid precursor protein (APP) [6], constituting a toxic peptide that causes inhibition of long-term hippocampal potentiation (LTP) and reactive oxygen species (ROS) production [7] and promote neuronal death [8]. On the other hand, NFTs comprise the phosphorylated microtubule-associated protein tau and disturb the normal function of microtubules and neurons [9].

Glutamatergic signalling, which plays an important role in synaptic plasticity, is known to be dysregulated in AD, as neuronal hyperexcitability caused by suppression of glutamate reuptake has been described to be the earliest hallmark of AD in humans and animal models [10]. This glutamate spillover might overactivate N-methyl-D-aspartate receptors (NMDAR), which are heteromeric glutamate-gated ion channels implicated in synaptic plasticity, learning, and memory but also in neurodegeneration and excitotoxicity [11,12,13]. NMDARs act as non-selective cation channels that allow a voltage-dependent influx of calcium into the neurons [14]. Calcium is a major second messenger for neuronal synaptic functions and nervous system development, and dysregulation of calcium signalling has been found to be closely related to AD progression [15]. Actually, memantine, a non-competitive NMDA antagonist, is used clinically in the treatment of AD to address NMDAR excessive activation [16,17]. Furthermore, soluble oligomeric Aβ, which induces deterioration of synaptic function in Alzheimer’s disease even before signs of dementia and plaque formation appear [18], has been suggested to act through NMDAR [19].

NMDARs are tetramers composed of different subunits, and, up to now, seven different subunits have been identified: the GluN1 subunit, four distinct GluN2 subunits (NR2A, NR2B, GluN2C and GluN2D), which are encoded by four different genes, and a pair of GluN3 subunits (GluN3A and GluN3B) arising from two separate genes [14]. The existence of this large repertoire of homologous NMDAR subunits allows for various combinations of subunit assembly, which gives rise to a multiplicity of receptor subtypes in the CNS with different biophysical and pharmacological properties, interacting partners and subcellular localisations. Subunit composition differs across CNS regions, along developmental stages and in disease states [20,21,22]. NMDAR tetramers are typically composed of two obligatory GluN1 subunits together with two copies of GluN2 and/or GluN3 subunits [14].

NR2A and NR2B subunits differ in their spatiotemporal expression. In rodents, NR2B is widely expressed in the embryonic brain, while NR2A expression starts shortly after birth. Then, NR2A expression increases progressively and becomes abundant throughout the central nervous system [23,24,25]. The replacement of NR2B with NR2A subunits is not absolute, as NR2B subunits are still expressed in many regions of the adult CNS [14,25].

NMDA receptors are usually located at postsynaptic sites in dendritic spines, although NMDAR expression has also been found at presynaptic and perisynaptic sites [12]. Within individual neurons, different NMDAR subtypes are expressed, and they are segregated in an input-specific manner [26,27]. NMDAR subtypes also vary according to subcellular localisation, NR2A subunits appear to be more abundant in synaptic sites, while NR2B is present in both synaptic and extrasynaptic domains [12,28]. Moreover, NMDARs are highly mobile at the membrane and probably exchange through lateral diffusion between synaptic and extrasynaptic sites [29]. NR2B-containing receptors have faster rates of diffusion than NR2A-containing receptors, which contributes to the enrichment of NR2A at mature synaptic sites [30].

NR2A and NR2B subunits also have different gating and kinetic properties; thus, subunit content affects NMDAR biophysical, pharmacological and signalling attributes. When agonists are fully bound to the receptors GluN1/NR2A, receptors have a higher open probability than GluN1/NR2B [22]. In pyramidal cells, synaptic NMDARs that usually contain NR2A subunits show faster kinetics, whereas extrasynaptic NMDARs, mainly containing NR2B subunits, display slower kinetics [31]. Therefore, the synaptic NR2B/NR2A ratio determines the consequences of NMDAR activation, including total calcium influx and downstream signalling. This ratio is not static, changing not only in response to neuronal activity and sensory experience during postnatal development, but also in adult synapses [14,32,33].

It is known that activation of synaptic NMDAR promotes ERK phosphorylation and activates CREB, facilitating the expression of plasticity-related genes [12], which have an important role in learning and memory [34,35]. On the contrary, the activation of extrasynaptic NMDAR leads to sustained dephosphorylation of CREB, rendering CREB transcriptionally inactive [36], which causes cell death and neurodegeneration [19]. Thus, the activation of NR2A-containing receptors is thought to promote neuronal survival, while NR2B activation can be related to the pathological progress of AD. Actually, aberrant and synergistic activation of NR2B-containing NMDAR at extrasynaptic sites by glutamate and Aβ has been shown in AD [11,37,38]. Moreover, both Aβ [39] and tau [40] have been shown to alter synaptic functions by removing synaptic NMDARs. Thus, the deleterious effects of Aβ prior to neuronal loss seem to be mediated by NMDARs, especially NR2B-NMDARs [19,41].

In this sense, the aim of the present study consists of the analysis of the functional differences between NR2A or NR2B-containing NMDAR in primary neurons treated with Aβ and to describe their involvement in synaptic plasticity and Aβ axonal transport.

## 2. Results

### 2.1. Activation of NR2A-Containing NMDAR Shows Higher Levels of Calcium Release and MAPK Phosphorylation than NR2B-Containing NMDAR

NMDAR is a tetrameric structure formed by different combinations of subunits that constitute a multiplicity of receptor subtypes in the CNS with different biophysical and pharmacological properties, interacting partners and subcellular localisations. The most common NMDAR structures in humans are constituted of two GluN1 (NR1) subunits and two NR2A (NR2A) or two NR2B (NR2B) subunits. To go deep inside the functional differences between both structures, HEK-293T cells were transfected with the cDNAs for NR1 and NR2A fused to the Renilla luciferase protein (NR2A-Rluc) or the NR2B fused to the Renilla luciferase protein (NR2B-Rluc), treated with increasing concentrations of NMDA (from 100 nM to 300 µM) and then the calcium release and MAPK phosphorylation were analysed. The receptor expression was quantified by the analysis of Rluc emission, ensuring similar values between NR2A and NR2B (between 100,000 and 120,000 bioluminescent arbitrary units). As observed in Figure 1, stimulation of NR2A-containing NMDAR (Figure 1A,D) induced a stronger calcium release than NR2B-containing NMDAR (Figure 1B,D) (at 10 µM NMDA, fluorescent values are 900,000 AU versus 30,000 AU when comparing NR2A or NR2B-containing NMDAR, respectively). Similar results were obtained in MAPK phosphorylation, where NR2A-containing NMDAR induce a 310% increase in ERK1/2 phosphorylation while NR2B only induce a maximal effect of around 200% (Figure 1C,E).

### 2.2. NMDA-Induced Toxicity in Astroglia Cells Is Due to NR2B-Containing NMDAR

Recently, it has been described that NMDAR is expressed in astroglia, where overexposure to NMDA decreases the ability of cultured astrocytes to express glutamine synthetase, aquaporin-4 and the inward rectifying potassium channel Kir4.1, dysregulating astrocytes homeostatic function [42]. However, the underlying mechanism remains unknown. To evaluate any differences between NR2A and NR2B subunits of NMDA, mice primary cultures of astroglial cells were transfected with siRNA for NR2A or NR2B, to knock down NR2A and NR2B expressions, respectively. Western blot assay was performed using specific antibodies to ensure a decrease stronger than 70% (Figure 2D). These cells were treated with increasing concentrations of NMDA (from 10 nM to 300 µM) and MAPK phosphorylation was evaluated. In Figure 2A, a significant increase in MAPK phosphorylation (around 200% increase over basal levels) in control astroglia can be observed. Similar results were observed in the absence of NR2B; however, the NR2A KD showed a lower signal (around 150% increase over basal levels). These results agree with those obtained in transfected HEK-293T cells, where the NR2A-containing NMDAR showed stronger activity.

The same experiment was repeated in astroglia activated by pre-treatment with 1 μM LPS plus 200 U/mL IFN-γ for 48 h. The results in Figure 2B indicate a stronger activation of NMDAR induced by NMDA in activated astroglia compared to resting cells (around 300% increase over basal levels). Moreover, NR2A KD showed lower effects than NR2B KD, indicating that also in activated astroglia NMDA induces a stronger activity through NMDAR containing the NR2A subunit.

Finally, astroglia survival was evaluated after 48 h of treatment with 1 μM LPS plus 200 U/mL IFN-γ. Interestingly, it was observed that astroglia KD for NR2A presented a 30% decrease in cell survival, while astroglia KD for NR2B, thus expressing NR2A, showed similar results as control astroglia (Figure 2C). Lower but similar results were observed when astroglia was treated with NMDA 15 µM.

### 2.3. NMDA-Induced Cytotoxicity Derives from Stimulation of NR2B-Containing NMDAR 

Overactivation of NMDA receptors in neuronal cells induces cytotoxicity and neuronal death. To relate this effect to NR2A or NR2B subunits of NMDA, cortical and hippocampal primary cultures of neurons, the most affected areas in Alzheimer’s disease, were prepared. First, a RT-PCR assay was performed using specific primers to bind NR1A, NR2A or NR2B subunits of NMDAR. A stronger expression of NMDAR was observed in the hippocampus compared to the cortex (Figure 3A,B). Then, cortical and hippocampal neuronal primary cultures were transfected with siRNA for NR2A or NR2B to knock down NR2A and NR2B expression, respectively. Western blot assay was performed using specific antibodies to ensure a decrease stronger than 80% (Figure 3G,H). Different studies demonstrate that soluble Aβ oligomers have an important role in the onset of synaptic dysfunction in the early stages of AD. Moreover, Aβ1-42 is one of the most cytotoxic oligomers [19,43]. Then, when primary cultures of cortical neurons were pre-treated with Aβ1-42 (1 µM) for 48 h, a significant increase in NR1 subunit expression was observed, which was stronger in the NR2A subunit (Figure 3A). Similar results were obtained in the hippocampus (Figure 3B).

Then, neuronal survival was analysed after NMDA (15 µM) stimulation in the control, NR2A KD and NR2B KD cortical neurons. First, a 35% decrease in neuronal survival in control cells treated with NMDA was observed (Figure 3C,D). Similar results were obtained in NR2A KD neurons (blue bars) (Figure 3E,F). However, NR2B KD neurons (violet bars) showed similar survival when comparing NMDA to vehicle stimulations (Figure 3E,F). These results indicate that NMDA cytotoxicity is due to NMDA binding to the NR2B-containing NMDAR.

To move closer to Alzheimer’s pathology, neuronal survival after Aβ1-42 (1 µM) treatment was analysed. Results indicate that control cells and NR2A KD cells suffer an important reduction in cell survival 48 h after treatment (42% and 65% of cell survival, respectively), which was slighter in NR2B KD cells (Figure 3C–F). Thus, toxicity induced by Aβ1-42 is more forceful in cells expressing the NR2B-containing NMDAR. Similar results were obtained in hippocampal neurons (Figure 3D,F).

Another way to assess neuronal death relies on the analysis of different immunocytochemical markers. In this sense, hippocampal primary cultures of neurons KD for NR2A or KD for NR2B were treated with Aβ1-42 (1 µM) for 48 h or vehicle and stained with NISSL, KI-67 and caspase-3.

NISSL staining is redistributed in the cell body of injured or regenerating neurons, indicating the pathophysiological state of the neuron. The results observed in Figure 4A indicate an increase in NISSL staining after Aβ1-42 treatment. Interestingly, the fluorescent signal shows a higher increase in neurons KD for NR2A, while neurons KD for NR2B show similar levels to control neurons. This result indicates a correlation between NR2B-containing NMDAR and injured neurons due to Aβ1-42 treatment. Then, the NR2B subunit could be directly or indirectly involved in Aβ1-42 induced cytotoxic effects.

Ki-67 is a biomarker of cell proliferation. In this sense, it was observed that Aβ1-42 treatment induces a slight decrease in hippocampal neuronal proliferation. However, in neurons KD for NR2A, this decrease is higher, while in neurons KD for NR2B, the decrease is not observed. This result indicates that the NR2A subunit correlates with a better physiological condition of neurons (Figure 4B). Caspase-3 is a member of a conserved family of proteins involved in apoptosis. However, there is growing evidence indicating that caspase-3 is involved in regulating the growth and homeostasis of neurons [44]. When analysing the caspase-3 marker, it was observed that Aβ1-42 treatment induced a strong decrease in the fluorescence signal. This result is observed in control neurons, neurons KD for NR2A and neurons KD for NR2B. Nonetheless, the fluorescent signal is lower in KD neurons for NR2A than in control neurons (Figure 4C).

Altogether, these results point to the NR2B-containing NMDAR as responsible for NMDAR cytotoxic effects due to NMDA treatment, favouring the neurodegeneration induced by Aβ1-42.

### 2.4. Aβ1-42 Treatment Potentiates NR2B-Containing NMDAR but Not NR2A-Containing NMDAR in Primary Neurons

NMDAR is an ionotropic channel that when activated induces calcium release but also MAPK phosphorylation. To go deep inside NMDAR signalling, neuronal primary cultures of the cortex and hippocampus were prepared and treated with Aβ1-42 (1 µM) for 48 h. When analysing MAPK phosphorylation, stronger signalling was observed when neurons were treated with Aβ1-42 (Figure 5A,B). The increase observed in NMDAR expression after the Aβ1-42 treatment (Figure 3A,B) could explain the stronger MAPK signalling detected after the Aβ1-42 treatment. Then, the same experiment was performed in primary cultures of neurons KD for NR2A or NR2B subunits. In agreement with the results obtained in transfected HEK-293T cells, the NR2A-containing NMDAR shows stronger signalling than NMDAR containing NR2B subunit. Moreover, Aβ1-42 treatment potentiated NR2B-containing NMDAR signalling in cortical and hippocampal neurons, but not NR2A-containing NMDAR (Figure 5C,D).

In calcium release assays, Aβ1-42 treatment of neuronal primary cultures induced a stronger signal in comparision to control cells (Figure 5E,F) in the cortex and hippocampus. Moreover, neurons KD for NR2B showed stronger signalling than neurons KD for NR2A. Also, Aβ1-42 treatment showed a strong increase in calcium release in neurons KD for NR2A, but only a slight increase in neurons KD for NR2B (Figure 5G,H).

### 2.5. NMDAR Activation Does Not Affect Aβ1-42 Axonal Transport 

Alzheimer’s disease is a neurodegenerative pathology with an unknown cause and only symptomatologic treatments [45]. An interesting way to protect neurons from death relies on reducing Aβ1-42 transport, and thus the spread of the pathology to healthy neurons. Microfluidic standard neuronal devices contain 150 µm long channels that allow axonal terminals–dendrites contact between neurons of different wells avoiding cell body contact. These devices have become a new methodology to study protein aggregates transport. Then, primary cultures of control, KD for NR2A or KD for NR2B cortical neurons were seeded in microfluidic devices for 10 days until axons fully crossed the microgrooves, and were subsequently treated with Aβ1-42 for 48 h. Then, Aβ1-42 was stained with a selective antibody over Hoechst-stained nuclei. Results in Figure 6A–C indicate a significant Aβ1-42 axonal transport in control neurons. This Aβ1-42 transport was similar in neurons KD for NR2A and neurons KD for NR2B. Then, NMDA subunits do not differentially affect Aβ1-42 axonal transport.

### 2.6. NR2A-Containing NMDAR Favours Neurite Formation in Comparison to NR2B-Containing NMDAR

NMDAR activation is directly related to neuronal plasticity and thus to neurite formation. Then, primary cultures of cortical neurons were transfected to generate NR2A KD and NR2B KD, fixed and stained to detect neurite formation as red dots. As observed in Figure 6D, control neurons showed more neurite formation than NR2B KD neurons. However, NR2B KD neurons showed more neurite formation than NR2A KD.

On the other hand, when control primary neurons were treated with Aβ1-42 and NMDA for 48 h, an important decrease in neurite formation was observed. This decrease was stronger in neurons KD for NR2A and slighter in neurons KD for NR2B (Figure 6D). When the same experiment was performed in hippocampal neurons, similar results were obtained (Figure 6E). Figure 6F shows representative images of neurite quantification in hippocampal neuronal cultures.

Consequently, it seems that the expression of NR2A-containing NMDAR in neurons shows a protective role in front of Aβ1-42 toxicity, which can be detected by examining neurite formation. On the contrary, neurons expressing NR2B-containing NMDAR are more sensitive to Aβ1-42 and NMDA-induced cytotoxicity, and show a higher neuronal death.

## 3. Discussion

Upon aging, NMDAR-dependent processes become dysregulated, resulting in alterations in LTP and LTD [46], elevated postsynaptic calcium levels [47] and memory deficits [48]. Thus, NMDAR gain of function has emerged as a possible explanation for age-related synaptic impairments [49]. It has been suggested that Aβ accumulation may activate NMDARs at early stages of AD [50]. Actually, NMDAR dysregulation evoked by Aβ and the consequent loss of Ca^2+^ homeostasis is thought to be related to the early cognitive deficits observed in AD. Early synaptic dysfunction in AD is associated with increased oligomeric Aβ1-42, which has been described to cause NMDAR-dependent LTD and spine elimination [51,52], together with an impairment of LTP [53]. Moreover, the NMDAR antagonist D-VAP was able to block the Aβ-induced depression of glutamatergic transmission [50,54].

NMDAR can present different properties depending on their subunit composition. Extracellular Aβ has been reported to colocalise with NR2B subunits of NMDA receptors but not with NMDA NR2A subunits [55], and Aβ1-42 accumulation is known to activate NR2B-NMDA receptors [56]. These data, together with the fact that NR2B subunits show an inverse correlation with memory performance [57], indicate that NR2B subunits, but not NR2A, could have a deleterious effect on AD.

To afford new data on differences between NR2A or NR2B-containing NMDAR activity was first analysed in transfected HEK-293T cells. In agreement with the fact that agonists fully bound to GluN1/NR2A NMDAR have a higher open probability than GluN1/NR2B NMDAR [22], a more potent effect was observed when activating NR2A-containing NMDAR in comparison to NR2B-containing NMDAR, both in MAPK phosphorylation and calcium release.

Afterwards, it was demonstrated that NR2A-containing NMDAR also shows stronger signalling than NR2B-containing NMDAR in primary cultures of cortical and hippocampal neurons by analysing calcium release and MAPK phosphorylation. Interestingly, pre-treatment with Aβ1-42 strongly potentiated NR2B-containing NMDAR signalling, while it did not affect receptors expressing NR2A. These results demonstrate how amyloid beta peptide 1–42 disturbs intracellular calcium homeostasis through activation of NR2B-containing N-methyl-d-aspartate receptors in cortical cultures [58,59] and also in hippocampal cultures, while slightly affecting the NR2A subunit. In the same line, it has been observed that Ca^2+^ level mobilisation caused by Aβ1-42 was counteracted by NR2B selective antagonist ifenprodil [58]. Conversely, NR2A-NMDAR antagonism potentiated Ca^2+^ rise induced by a high concentration of Aβ1-42 (1 µM), suggesting that NR2A and NR2B subunits have opposite roles in regulating Ca^2+^ homeostasis [58]. Aβ accumulation was also described to enhance CA1 neuronal excitability through NR2B-NMDA receptors [56].

Another important point consisted in the evaluation of neuronal death after Aβ1-42 treatment, as inhibiting the over-activation of NR2B-containing NMDAR has been shown to be involved in the neuroprotective effect of silibinin on STZ-induced sporadic AD models [60]. In this sense, we observed that neurons expressing NR2A-containing NMDAR suffered less neuronal death after Aβ1-42 treatment than neurons expressing the NR2B subunit. Also, NMDA treatment induced less cytotoxicity in neurons containing NR2A than NR2B. These results agree with the demonstration that the antagonist PEAQX (for NR2A-containing NMDARs) did not prevent synaptotoxicity [61].

Aβ oligomers not only have been described to potentiate the NR2B-containing NMDAR, but also to inhibit LTP (Li et al., 2011) and facilitate LTD [52] by activation of extrasynaptic NR2B-containing receptors, thereby impairing synaptic plasticity and contributing to spine loss. Both LTP impairment and LTD facilitation were prevented by antagonists that are selective for NR2B-NMDARs [41,52,62,63,64,65]. NR2B subunit-containing NMDA receptor antagonists also prevented Aβ-mediated synaptic plasticity disruption in vivo [64].

Several studies demonstrate that NR2B subunits mediate the effects of Aβ on synapse stability. NR2B-NMDAR antagonist ifenprodil was able to prevent the synapse loss induced by incubation with exogenous Aβ in hippocampal cell cultures [41], indicating that NR2B-containing NMDARs play a central role in the pathology of Aβ neurotoxicity. Aβ was also described to cause loss of synaptic proteins PSD-95 and synaptophysin by NR2B-containing NMDAR activation, accompanied by the suppression of NR2A-containing NMDAR function [66]. However, Tackenberg and collaborators described that Aβ induces dendritic spine loss via a pathway involving NR2A-containing NMDARs, whereas activation of extrasynaptic NR2B-containing NMDARs is required for pTau-dependent neurodegeneration [61]. Providing new data on this controversy, we have observed that neurons expressing NR2B-containing NMDAR show an important decrease in neurite formation after Aβ1-42 treatment that has not been observed in neurons expressing NR2A-containing NMDAR.

Further evidence of the detrimental relationship between Aβ and NR2B subunits is the fact that the production of Aβ requires activation of extrasynaptic NMDAR [37,56]. Additional evidence that NR2B subunits are related to AD hallmarks is the finding that activation of extrasynaptic NMDA receptors induces tau overexpression and phosphorylation, forming neurofibrillary tangles [67]. Also, Aβ interaction with NR2B-NMDAR downregulates GluN1 mRNA levels [68]. We questioned if NMDAR activation could affect pTau axonal transport between cortical neurons. However, results indicated that neither NR2A-containing NMDAR nor NR2B-containing NMDAR were involved in pTau spread through cortical neurons, and then somehow related to AD progression.

On the contrary, activation of synaptic NR2A-containing NMDAR has been reported to have beneficial effects in an AD context. It has been described that synaptic NMDA receptor activation stimulates alpha-secretase amyloid precursor protein processing and inhibits amyloid-beta production, indicating that calcium influx through synaptic NMDA receptors promotes nonamyloidogenic alpha-secretase-mediated APP processing [69]. Also, we have observed that Aβ treatment does not alter NR2A-containing NMDAR signalling, while it protects from neuronal death after Aβ1-42 stimulation. Moreover, treatment with GNE-0723, a positive allosteric modulator of NR2A, reduces aberrant low-frequency oscillations and epileptiform discharges and improves cognitive functions in AD mouse models, suggesting that NR2A-subunit-containing NMDAR enhancers may have therapeutic benefits in brain disorders with network hypersynchrony and cognitive impairment [70]. 

Taken together, our results indicate that NMDA receptors containing NR2B subunit are dysregulated by Aβ1-42 oligomers, resulting in disruption of glutamatergic synaptic transmission, which parallels early cognitive deficits, while NMDA receptors containing NR2A seem to induce a protective role. Then, Aβ1-42 appears to shift the balance between synaptic and extrasynaptic NMDA receptors signalling towards the extrasynaptic receptor-mediating neurotoxicity.

## 4. Materials and Methods

### 4.1. Reagents

Lipopolysaccharide (LPS) and interferon-γ (IFN-γ) were purchased from Sigma-Aldrich (St Louis, MO, USA), NMDA and MK-801 from Tocris Bioscience (Bristol, UK) and Human Ab-oligomers from Anaspec (Fremont, CA, USA) (Cat. No. AS-20276). The human versions of NMDAR subunits NR1, NR2A and NR2B were obtained from Addgene (Watertown, MA, USA) and subcloned to pcDNA3.1. The antibodies used were the following: polyclonal rabbit anti-Nissl (Life technologies, Carlsbad, CA, USA, N21480), Monoclonal mouse anti-Ki-67 (14-5698-82, Invitrogen, Waltham, MA, USA) and polyclonal rabbit anti-Caspase3 (9H19L2, Thermo Fischer Scientific, Waltham, MA, USA), polyclonal rabbit anti-Nectin 3 (ab63931, Abcam, Cambridge, UK) and polyclonal rabbit anti-F-actin antibody fused to an Alexa 488 fluorophore (A12379, ThermoFisher, Waltham, MA, USA). pTau protein was kindly provided by Prof. J. Avila (CBM, UAM-CSIC, Madrid, Spain).

### 4.2. Aβ-Oligomer Production

Human Ab-oligomers (Cat. No. AS-20276 Anaspec) were prepared according to a previously established protocol [10.15252/embr.201643519]. Briefly, the lyophilised Aβ1-42 was disaggregated in 1,1,1,3,3,3-hexafluoro-2-propanol (HFIP, Sigma-Aldrich, St Louis, MO, USA, Cat. No. B2517) to 0.5 mg/mL. The solution was aliquoted. After the evaporation of HFIP, the peptide film was stored at −80 °C. Twenty-four hours before use, the peptide film was dissolved in DMSO (1:1000, Cat. No. 276855, Sigma-Aldrich, St Louis, MO, USA), sonicated, diluted to 50 µM concentration in F12 medium (Cat. No. 21765-029, Gibco, Seoul, Republic of Korea) and kept for oligomerisation at 4 °C for 24 h. For every independent experiment, a fresh preparation of oligomers was used.

### 4.3. Cell Culture and Transfection

HEK-293T cells were grown in Dulbecco’s modified Eagle’s medium (DMEM) supplemented with 2 mM L-glutamine, 100 U/mL penicillin/streptomycin, and 5% (*v*/*v*) heat-inactivated Fetal Bovine Serum (FBS) (Invitrogen, Paisley, Scotland, UK). Cells were maintained in a humid atmosphere of 5% CO_2_ at 37 °C. Cells were transiently transfected with the PEI (PolyEthylenImine, Sigma, St. Louis, MO, USA) method as previously described [71]. Briefly, the corresponding cDNAs diluted in 150 mM NaCl were mixed with PEI (5.5 mM in nitrogen residues) also prepared in 150 mM NaCl for 10 min. The cDNA-PEI complexes were transferred to HEK-293T cells and were incubated for 4 h in a serum-starved medium. Then, the medium was replaced by a fresh supplemented culture medium, and cells were maintained at 37 °C in a humid atmosphere of 5% CO_2_ for 48 h.

### 4.4. Primary Cultures

Astroglial primary cultures were prepared from the cortex of 2-day C57/BL6 mice. In brief, samples were dissected, carefully stripped of their meninges, and digested with 0.25% trypsin for 30 min at 37 °C. Cells were brought to a single-cell suspension by repeated pipetting followed by passage through 100 µm pore mesh and pelleted (7 min, 200 g). Astroglia were resuspended in supplemented DMEM medium (2 mM L-glutamine, 100 U/mL penicillin/streptomycin, and 5% (*v*/*v*) heat-inactivated Fetal Bovine Serum (FBS) (Invitrogen, Paisley, Scotland, UK)) and seeded at a density of 200,000 cells/cm^2^ in 6-well plates. Cultures were maintained at 37 °C in a humidified 5% CO_2_ atmosphere for 12 days.

Neuronal primary cultures were isolated as previously described [72]. Briefly, cultures were prepared from the cortex and hippocampus of fetuses from 18–19 days of C57/BL6 pregnant mice. Samples were dissected, carefully stripped of their meninges, and digested with 0.25% trypsin for 30 min at 37 °C. Cells were brought to a single-cell suspension by repeated pipetting (0.5 to 0.8 mm nails) followed by passage through 100 µm pore mesh and pelleted (7 min, 200 g). Neurons were resuspended in supplemented DMEM medium (2 mM L-glutamine, 100 U/mL penicillin/streptomycin, and 5% (*v*/*v*) heat-inactivated Fetal Bovine Serum (FBS) (Invitrogen, Paisley, Scotland, UK)) and seeded at a density of 400,000 cells/mL in 6-well plates. Then, 24 h after, the medium was exchanged to neurobasal medium supplemented with 2 mM L-glutamine, 5% (*v*/*v*) FBS, 100 U/mL penicillin/streptomycin, and 2% (*v*/*v*) B27 supplement (Gibco) and cultures were maintained at 37 °C in humidified 5% CO_2_ atmosphere for 12 days.

### 4.5. Knockdown of Endogenous NR2A or NR2B Subunits of NMDAR in Primary Cultures of Mice Neurons

Primary striatal neurons growing in 6-well dishes were transfected with the Lipofectamine method to knock down the NR2A or the NR2B expression using siRNAs (Merck, Darmstadt, Germany). Briefly, the corresponding siRNAs (1 µg/well) diluted in 50 µL of non-supplemented DMEM were mixed with 1 µL lipofectamine per each µL siRNA also prepared in 50 µL of non-supplemented DMEM. The cDNA–lipofectamine complexes formed during 20 min were transferred to neuronal or astroglial primary cultures and were incubated for 8 h in the serum-starved medium. After, the medium was replaced with a complete culture medium. Forty-eight hours after transfection, neurons were then detached and NR2A and NR2B expression was detected by Western blotting.

Briefly, the cell lysis was performed by the addition of 250 µL of ice-cold lysis buffer. Cellular debris was removed by centrifugation at 13,000× *g* for 5 min at 4 °C, and protein was quantified by the bicinchoninic acid method using bovine serum albumin dilutions as standard. To determine the level of NR2A and NR2B subunits, equivalent amounts of protein (10 μg) were separated by electrophoresis on a denaturing 10% SDS-polyacrylamide gel and transferred onto PVDF-FL membranes. Membranes were blocked with Odyssey blocking buffer (LI-COR Biosciences, Lincoln, Nebraska) for 60 min and probed with a mixture of a mouse anti-phospho-ERK1/2 antibody (1:2500, Sigma) and rabbit anti-ERK1/2 antibody (1:40,000, Sigma) for 2 h. The bands corresponding to NR2A and NR2B were visualised by the addition of a mixture of IRDye 800 (anti-mouse) antibody (1:10,000, Sigma) and IRDye 680 (anti-rabbit) antibody (1:10,000, Sigma) for 1 h and scanned by the Odyssey infrared scanner (LI-COR Biosciences). Band densities were quantified using the scanner software Imatge studio version 5.2. and exported to Microsoft Excel Version 2312. The levels of NR2A and NR2B were normalised for differences in loading using the total ERK1/2 protein band intensities.

The residual expression was lower than 20% [72].

### 4.6. Cell Viability

Cell viability assay is based on the principle that living cells maintain intact cell membranes that exclude certain dyes, like trypan blue. To quantify the percentage of living cells, cortical and hippocampal neurons or astroglia cells were plated in 6-well plates and kept in the incubator for 10 days. Then, cells were transfected with siRNA for NR2A, siRNA for NR2B or vehicle. After, neurons were treated with Aβ1-42 (1 µM) or vehicle for 48 h while astroglia were treated with 1 μM LPS plus 200 U/mL IFN-γ for 48 h.

Finally, cells were gently detached and mixed with an equal volume of trypan blue (0.4%). Cells (%) were counted with a TC20™ Automated Cell Counter (Biorad, 1450102).

### 4.7. Detection of Cytoplasmic Calcium Levels

HEK-293T cells were co-transfected with the cDNA for the protomers of the NMDA receptor NR1a (0.6 μg) and NR2B (0.6 μg), and with the cDNA for the GCaMP6 calcium sensor (0.5 μg) [10.1186/s13195-021-00920-6] by the use of the PEI method. Next, 48 h after transfection, the HEK-293T were incubated with Mg^2+^-free Locke’s buffer (154 mM NaCl, 5.6 mM KCl, 3.6 mM NaHCO_3_, 2.3 mM CaCl_2_, 5.6 mM glucose, 5 mM HEPES, 10 μM glycine, pH 7.4) and were plated in 96-well black, clear bottom plates. Online recordings were performed right after the addition of increasing concentrations of NMDA. When indicated cells were pre-treated with NMDAR antagonists MK-801 for 10 min. Fluorescence emission intensity due to complexes between GCaMP6 and Ca^2+^ was recorded at 515 nm upon excitation at 488 nm on the EnSpire^®^ Multimode Plate Reader for 150 s every 5 s at 100 flashes per well.

Primary cultures of neurons were seeded in transparent 96-well plates and maintained with supplemented neurobasal or supplemented DMEM, respectively, at 37 °C in a humidified 5% CO_2_ atmosphere for 10 days. Then, primary cultures were transfected with siRNA for NR2A, siRNA for NR2B or vehicle. Subsequently, neurons were treated with Aβ1-42 (1 µM) or vehicle for 48 h. After, cells were washed and incubated for 1 h with Mg^2+^-Ca^2+^-free Locke’s buffer (154 mM NaCl, 5.6 mM KCl, 3.6 mM NaHCO_3_, 5.6 mM glucose, 5 mM HEPES, 10 μM glycine, pH 7.4) and subsequently washed with Assay Buffer (Fluo-4 Direct™ Calcium Assay Kit (Thermo Fisher Scientific)). Cells were subsequently incubated for 1 h with Flu-4 reagent (Fluo-4 Direct™ Calcium Assay Kit (Thermo Fisher Scientific)). Finally, neurons were treated with increasing concentrations of NMDA, and, a few seconds later, fluorescence was read in a NIVO Multimode reader (Thermo Fisher).

### 4.8. ERK Phosphorylation Assays

To determine ERK1/2 phosphorylation, 40,000 transfected HEK-293T cells/well, 50,000 neurons/well or 40,000 astroglia cells/well were plated in transparent Deltalab 96-well microplates and kept in the incubator for 48 h (transfected HEK-293T cells) or 10 days (astroglial and neuronal primary cultures). Then, primary cultures were transfected with siRNA for NR2A, siRNA for NR2B or vehicle. Then, neurons were treated with Aβ1-42 (1 µM) or vehicle for 48 h while astroglia was treated with 1 μM LPS plus 200 U/mL IFN-γ for 48 h. Two to four hours before initiating the experiment, the medium was substituted by a serum-starved DMEM medium. Then, cells were pre-treated at 25 °C for 10 min with the specific antagonist MK-801 or vehicle in serum-starved DMEM medium and stimulated for an additional 7 min with increasing concentrations of NMDA or vehicle. Cells were then washed twice with cold PBS before the addition of lysis buffer (20 min treatment in constant agitation). An amount of 10 μL of each supernatant was placed in white ProxiPlate 384-well microplates and ERK 1/2 phosphorylation was determined using AlphaScreen^®^SureFire^®^ kit (Perkin Elmer) following the instructions of the supplier and using an EnSpire^®^ Multimode Plate Reader (PerkinElmer, Waltham, MA, USA).

### 4.9. Quantitative Polymerase Chain Reaction (qPCR)

The hippocampus and cortex were isolated from C57BL/6J mice at 4 months old. After decapitation, the hippocampus and cortex were dissected and immediately transferred to PBS. mRNAs were isolated with TRIzol™ Reagent and treated with RNase-Free DNase (Qiagen, Hilden, Germany). The purity was verified with a NanoDrop 2000 Spectrophotometer (Thermo Scientific). Single-strand cDNA was synthesised from the extracted RNA (2 μg) using a MLV-reverse transcriptase (Fisher Scientific), Random Hexamers and oligo-dT. The qPCR was conducted with the cDNA and PowerUp SYBR Green Master Mix (AppliedBiosystems, Waltham, MA, USA). Determinations were conducted using real-time PCR technique on an Applied Biosystems QuantStudio3 device in 96-well plates.

The primer pair for the gene that codifies for NR1 subunit were CAAGTATGCGGATGGGGTGA (forward) and CAGTCTGGTGGACATCTGGTA (reverse). The primer pair for the gene that codifies for NR2A subunit were CCCCAAACTCCTCAAATCAA (forward) and CAGGCGACTCAGAAATGACA (reverse). The primer pair for the gene that codifies for NR2B subunit was ATTGGTGGCAGAGTGGATTC (forward) and GGCAAAAGAATCATGGCTGT (reverse). Primers for beta-actin were used as an internal control (forward, TTGACATCCGTAAAGACCTC; reverse, AGGAGCCAGAGCAGTAAT). Each experiment included a template-free control. The PCR products were analysed by the DNA melting curve. The relative quantities codifying for NR1A, NR2A and NR2B PCR products were estimated with respect to the amount of the housekeeping gene beta-actin product using the ΔCt method: %beta-actin = (2Ct of beta-actin − Ct of NR1A/NR2A/NR2B) × 100.

### 4.10. Immunocytochemistry 

Primary hippocampal neuronal culture cells seeded in coverslips were fixed in 4% paraformaldehyde for 15 min and washed twice with PBS containing 20 mM glycine, before permeabilization with PBS-glycine containing 0.2% Triton X-100 (15 min). Neurons were treated for 1 h with PBS containing 1% BSA and labelled with the antibodies anti-Nissl (Life technologies, N21480), anti-Ki-67 (Invitrogen, 14-5698-82) and anti-Caspase3 (Thermo Fischer Scientific, 9H19L2). Samples were washed several times and mounted with 30% Mowiol. Samples were observed in a Leica SP2 confocal microscope (Leica Microsystems, Mannheim, Germany). Scale bar: 30 µm. Fluorescence was quantified by Fiji–Image J software (version 2.14.0) considering different fields and dividing the total fluorescence by the number of nuclei in each field.

### 4.11. Microfluidics Assays of Tau Trafficking

Microfluidic standard neuronal devices (with 150 µm microgroove barriers located in the area between the channels; AXIS™ AXon Investigation System, EMD Millipore, Burlington, MA, USA) were handled following the manufacturer’s protocol as previously described [73]. Briefly, each assembled device was coated with poly-D-lysine (0.1 mg/mL, Gibco A3890401) for 1 h. Then, the pure cortical neurons from mouse embryos (E19) were isolated and a total of 10 microliters of cell suspension (5–6 million cells/mL) was added to both chambers of each device by passive pumping. After 30 min of incubation at 37 °C to allow cell attachment, 200 µL of neurobasal medium (supplemented with 2 mM L-glutamine, 100 U/mL penicillin–streptomycin and 2% (*v*/*v*) B27 supplement (Gibco)) were added. Neurons were maintained at 37 °C in a humidified 5% CO_2_ atmosphere. The medium was replaced every three days in each device (a 50 µL difference in media volume was maintained to prevent spontaneous diffusion). After 10 days of incubation, neurons were transfected using the lipofectamine protocol with siRNA for NR2A, NR2B or vehicle. Next, 24 h after, once the axons fully crossed the microgrooves (150 µm distance) into the axonal compartment of a device, pTau was added into compartment A of each device. Then, after 24 h, to detect pTau protein, an immunostaining technique was applied using immunocytochemistry protocol. Neurons were labelled with rabbit anti-phospo-tau (S396) antibody (1/100, Abcam ab109390) and subsequently marked with a Cy3 anti-rabbit (1/200, Jackson InmunoResearch, West Grove, PA, USA) secondary antibody (red). Following 2 h of incubation, cells were washed and subsequently imaged after 24 h using a confocal microscope with 40× and 25× objectives (Zeiss LSM 880, Carl Zeiss, Jena, Germany).

### 4.12. Neurite Patterning Determination

Cortical and hippocampal neuronal primary cultures seeded for 10 days were transfected using the lipofectamine protocol with siRNA for NR2A, NR2B or vehicle. The next day, cells were treated with Aβ (500 nM) for 48 h and subsequently stimulated with NMDA (15 µM) or vehicle for 24 h. Then, cells were fixed in 4% paraformaldehyde for 15 min and then washed twice with PBS containing 20 mM glycine, followed by permeabilization with the same buffer containing 0.2% Triton X-100 (15 min incubation). Samples were treated for 1 h with blocking solution (PBS containing 1% bovine serum albumin) and labelled with polyclonal rabbit anti-Nectin 3 antibody (Abcam, 1/1000). Neurons were detected with 3 anti-F-actin antibodies fused to an Alexa 488 fluorophore (ThermoFisher, 1/400). Then, cells were incubated at RT for 2 h with a Cy3-conjugated anti-rabbit secondary antibody (1/200, 711-166-152, Jackson ImmunoResearch). Finally, samples were washed several times with PBS and mounted with 30% Mowiol (Calbiochem, San Diego, CA, USA). Nuclei were stained with Hoechst (1/100). Samples were observed under a Zeiss 880 confocal microscope (Leica Microsystems, Wetzlar, Germany). Quantification of neurite formation was performed over segments of 15 μm. Each red dot represents a neurite formation.

### 4.13. Data Analysis

The data in the graphs are the mean ± S.E.M. GraphPad Prism software version 5 (San Diego, CA, USA) was used for data fitting and statistical analysis. One- or two-way ANOVA followed by post hoc Bonferroni test was used depending on the number of factors. Two factors were considered in the case of ligand treatments in resting or activated astroglia (two levels) or in the case of ligand treatments in control or Aβ stimulated neurons (two levels). When a pair of values were compared, Student’s T test was used. Significant differences were considered when *p* < 0.05.

## Figures and Tables

**Figure 1 ijms-25-04757-f001:**
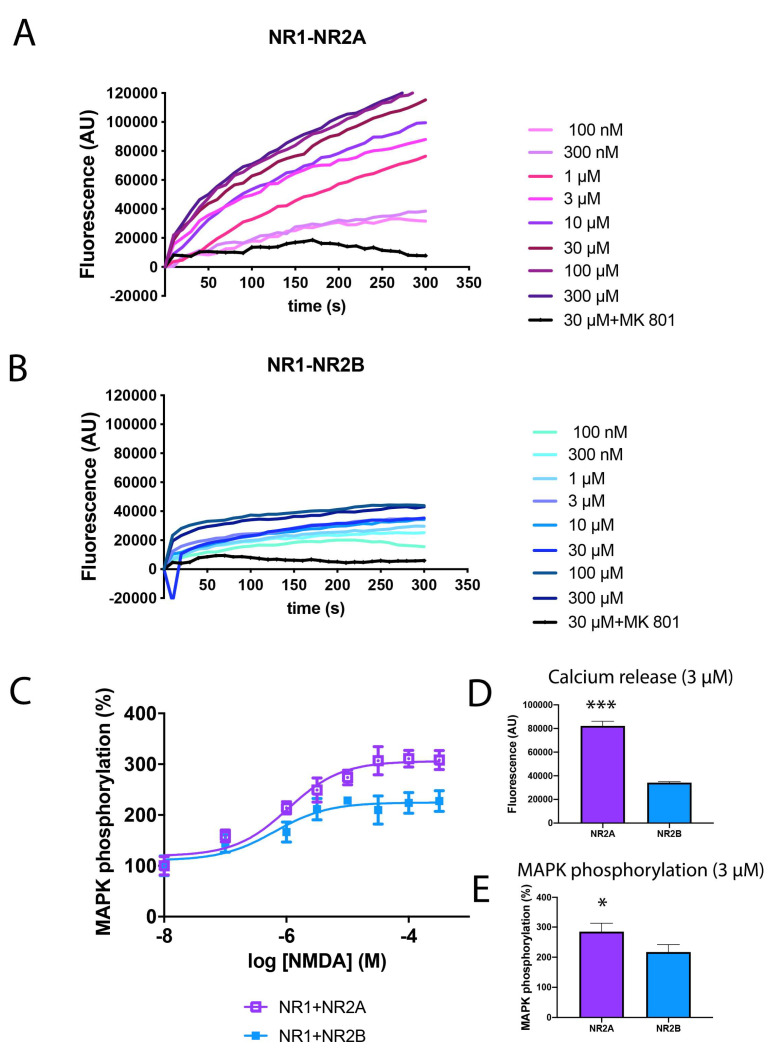
Differential signalling of NR2A or NR2B-containing NMDAR in transfected HEK-293T cells. (**A**,**B**,**D**) HEK-293T cells were transfected with the cDNAs for the NMDAR subunits NR1A (0.6 µg/well), NR2A (0.6 µg/well) (**A**) or NR2B (0.6 µg/well) (**B**) and the calcium sensor GCaMP6 (0.5 µg/well). Cells were pre-treated with MK-801 (1 µM) or vehicle and treated with increasing concentrations of NMDA (100 nM to 300 µM). Real-time calcium-induced fluorescence was collected for 300 s. Values are representative curves of 8 independent experiments performed in duplicates. (**C**,**E**) HEK-293T cells were transfected with the cDNAs NR1A (0.6 µg/well), NR2A (0.6 µg/well) (violet) or NR2B (0.6 µg/well) (blue). Cells were treated with increasing concentrations of NMDA (10 nM to 300 µM) and ERK1/2 phosphorylation was analysed using an AlphaScreen^®^SureFire^®^ kit (Perkin Elmer, Waltham, MA, USA). Values are the mean ± S.E.M. of 6 independent experiments performed in triplicates. (**D**,**E**) Calcium release and MAPK phosphorylation induced by 3 µM NMDA. Values are the mean ± S.E.M. of 8 independent experiments performed in duplicates. One-way ANOVA followed by Bonferroni’s multiple comparison post hoc test was used for statistical analysis (* *p* < 0.05, *** *p* < 0.001 comparing NR2A-containing NMDAR versus NR2B-containing NMDAR).

**Figure 2 ijms-25-04757-f002:**
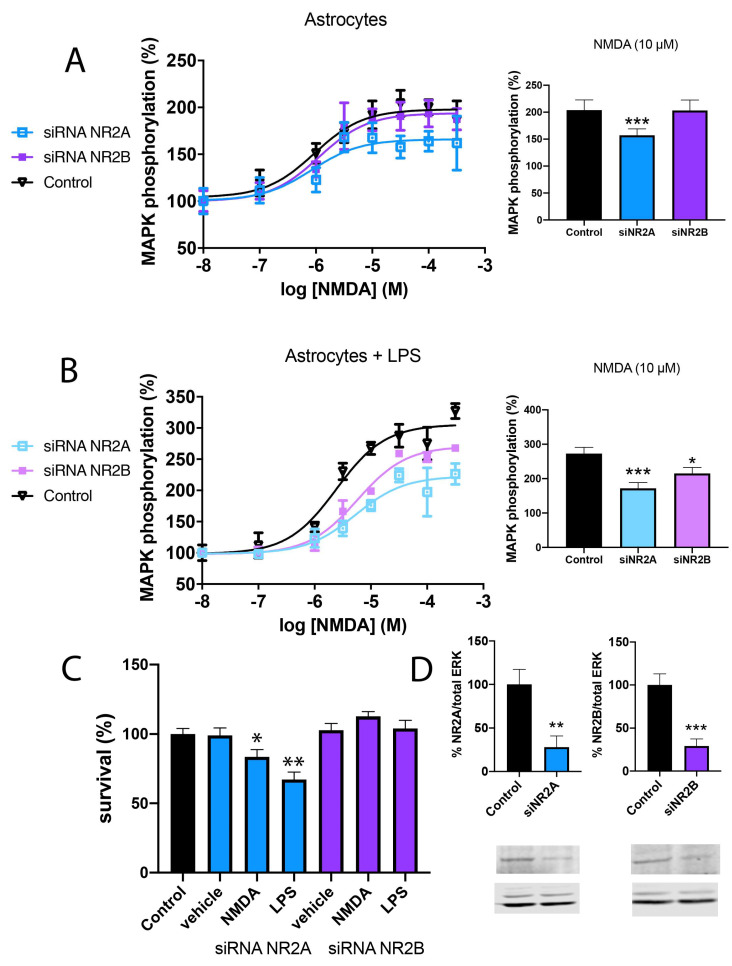
Differential signalling of NR2A or NR2B-containing NMDAR in primary cultures of astrocytes. Primary cultures of astrocytes were transfected with siRNA for NR2A (blue), siRNA for NR2B (violet) or vehicle (black) and treated with 1 μM LPS plus 200 U/mL IFN-γ (**B**,**C**) or vehicle (**A**) for 48 h. After, cells were treated with increasing concentrations of NMDA (10 nM to 300 µM) and ERK1/2 phosphorylation was analysed using an AlphaScreen^®^SureFire^®^ kit (Perkin Elmer) (**A**,**B**). Values are the mean ± S.E.M. of 6 independent experiments performed in triplicates. One-way ANOVA followed by Bonferroni’s multiple comparison post hoc test was used for statistical analysis (* *p* < 0.05, *** *p* < 0.001 versus control condition). Cells were treated with NMDA (15 µM) or vehicle for 24 h. Then, astroglia were gently detached and mixed with an equal volume of trypan blue (0.4%) and counted with a TC20™ Automated Cell Counter (1450102, Biorad, Hercules, CA, USA) (**C**). Values are the mean ± S.E.M. of 8 independent experiments performed in triplicates. One-way ANOVA followed by Bonferroni’s multiple comparison post hoc test was used for statistical analysis (* *p* < 0.05, ** *p* < 0.01 versus control condition). (**D**) Western blot of astroglia control, KD for NR2A or KD for NR2B. Expression was evaluated in percentage over control neurons and with respect to total ERK normalisation. Values are the mean ± S.E.M. of 5 independent experiments. Student’s *t* test was used for statistical analysis (** *p* < 0.01, *** *p* < 0.001 versus control condition).

**Figure 3 ijms-25-04757-f003:**
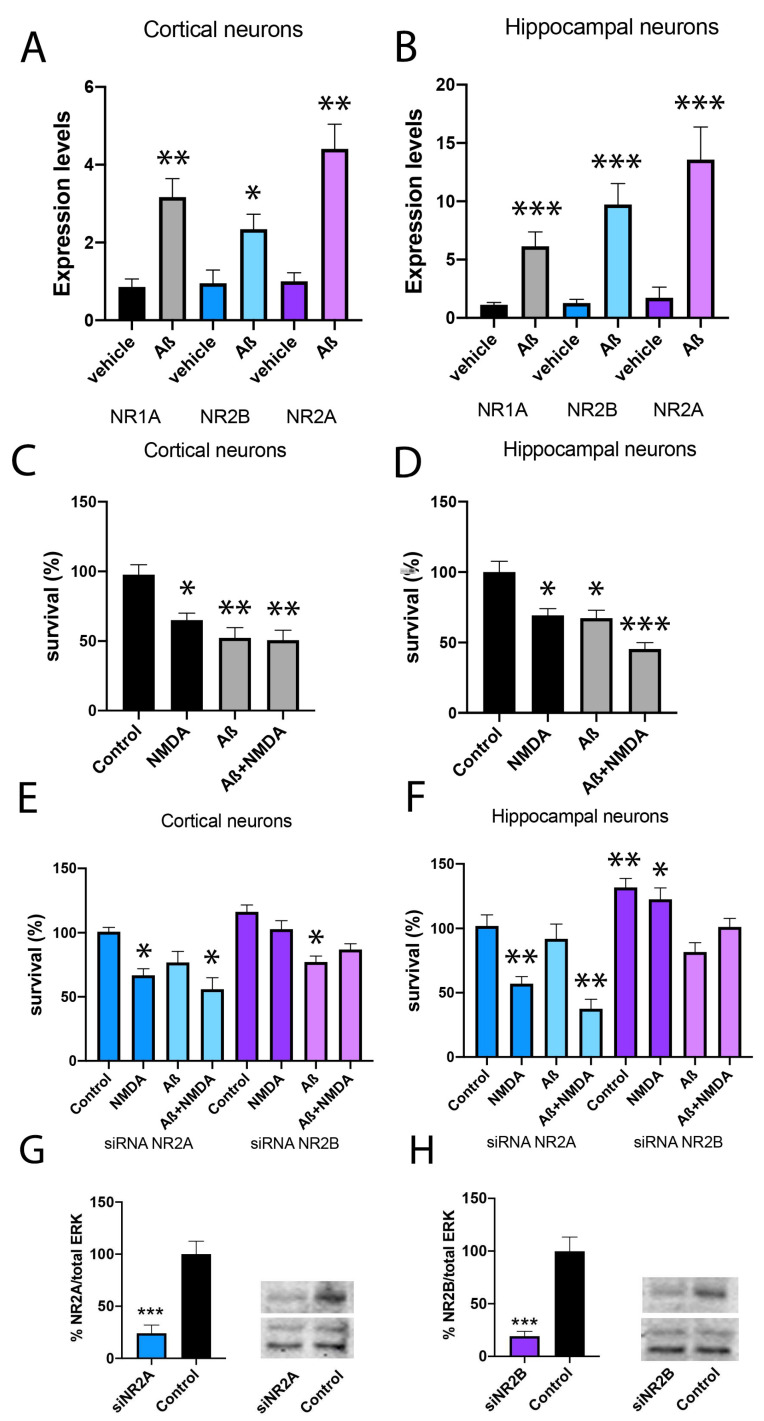
Analysis of neuronal survival in primary cultures silenced for NR2A or NR2B. (**A**,**B**) NR2A and NR2B expression levels were determined by RT-PCR in cortical and hippocampal primary cultures of neurons treated with Aβ1-42 (1 µM) or vehicle for 48 h. (**C**–**F**) Cortical and hippocampal primary cultures of neurons were transfected with siRNA for NR2A (blue), siRNA for NR2B (violet) or vehicle (black) and treated with Aβ1-42 (1 µM) or vehicle for 48 h followed by NMDA stimulation (15 mM) or vehicle for 24 h. Then, neurons were gently detached and mixed with an equal volume of trypan blue (0.4%) and counted with a TC20™ Automated Cell Counter (Biorad, 1450102). Values are the mean ± S.E.M. of 5 independent experiments performed in triplicates. One-way ANOVA followed by Bonferroni’s multiple comparison post hoc test was used for statistical analysis (* *p* < 0.05, ** *p* < 0.01, *** *p* < 0.001 versus control condition). (**G**,**H**) Western blot of cortical primary neurons control, KD for NR2A or KD for NR2B. Both NR2A and NR2B have molecular weights between 160 and 180 kDa. Expression was evaluated in percentage over control neurons and with respect to total ERK normalisation. Values are the mean ± S.E.M. of 6 independent experiments. Student’s *t* test was used for statistical analysis (*** *p* < 0.001 versus control condition).

**Figure 4 ijms-25-04757-f004:**
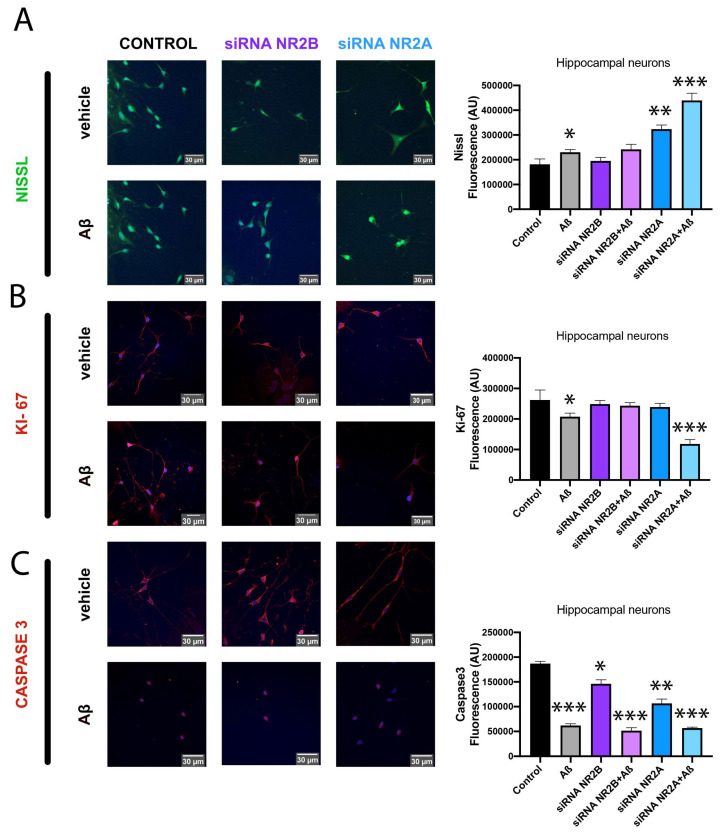
Immunocytochemistry assay of neuronal morphology in primary cultures treated with Aβ1-42. Immunocytochemistry assay was performed in hippocampal primary cultures of neurons transfected with siRNA for NR2A (blue), siRNA for NR2B (violet) or vehicle (black) and treated with Aβ1-42 (1 µM) or vehicle for 48 h. Neuronal survival was characterised by incubating with the specific antibodies: anti-NISSL (neuronal morphology) (**A**), anti-Ki-67 (neuronal proliferation) (**B**) or anti-Caspase3 (neuronal plasticity) (**C**). Detection is shown in green (**A**) or red (**B**,**C**) due to incubation with a secondary antibody conjugated to a fluorochrome Alexa 488 or Alexa 561, respectively. Scale bar: 30 µm. Values are the mean ± S.E.M. of 6 independent experiments performed in triplicates. One-way ANOVA followed by Bonferroni’s multiple comparison post hoc test was used for statistical analysis (* *p* < 0.05, ** *p* < 0.01, *** *p* < 0.001 versus control condition).

**Figure 5 ijms-25-04757-f005:**
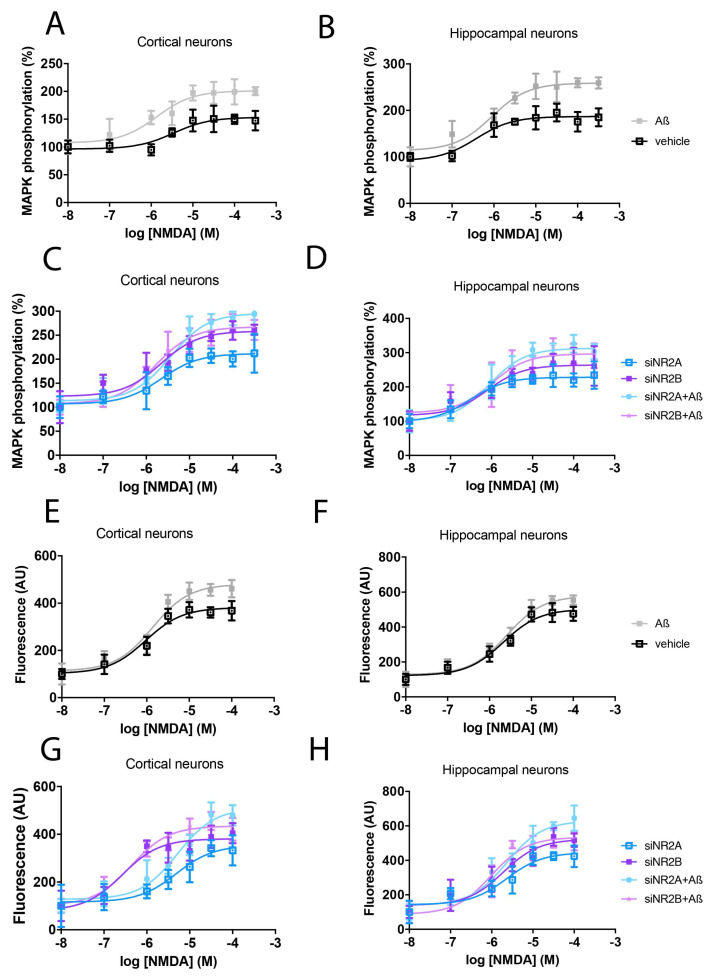
Differential signalling of NR2A or NR2B-containing NMDAR in primary cultures of neurons treated with Aβ1-42. Cortical and hippocampal primary cultures of neurons were transfected with siRNA for NR2A (blue), siRNA for NR2B (violet) or vehicle (black) and treated with Aβ1-42 (1 µM) or vehicle for 48 h. After, cells were stimulated with a range of NMDA concentrations (10 nM to 300 µM) or vehicle for 10 min in MAPK phosphorylation assays or a few seconds when detecting calcium release. ERK1/2 phosphorylation was analysed using an AlphaScreen^®^SureFire^®^ kit (Perkin Elmer) (**A**–**D**). Calcium release was detected by Fluo-4 Direct^TM^ Calcium Assay Kit (Thermo Fisher Scientific, Waltham, MA, USA) in a NIVO Multimode reader (Thermo Fisher) (**E**–**H**). Values are the mean ± S.E.M. of 8 independent experiments performed in triplicates.

**Figure 6 ijms-25-04757-f006:**
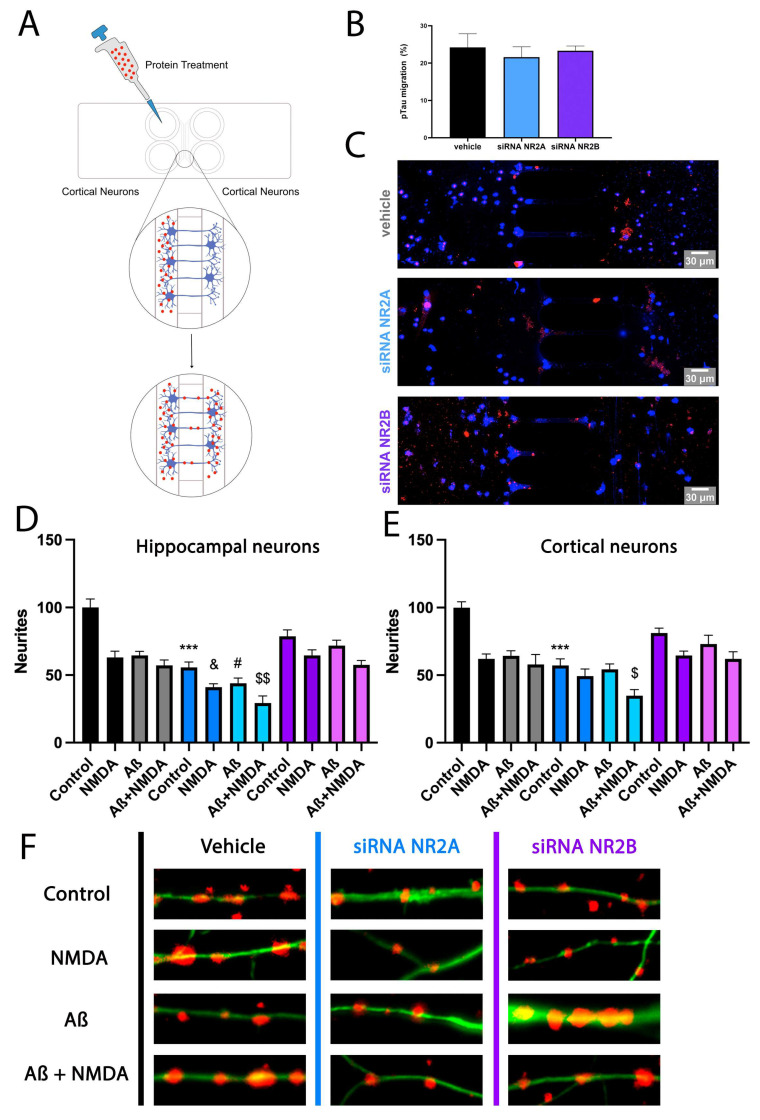
Effect of NR2A or NR2B-containing NMDAR in pTau migration and neurite patterning in primary cultures of neurons treated with Aβ1-42. (**A**–**C**) Mice cortical neurons were grown in microfluidic devices. On DIV 10, neurons were transfected with siRNA for NR2A (blue), siRNA for NR2B (violet) or vehicle (black). On DIV 11, pTau (1 µM) protein was added into compartment A of each device for 24 h. Neurons were labelled with anti-pTau antibody and subsequently with a Cy3 anti-rabbit antibody and imaged in a Zeiss LSM 880. Bar graph shows quantification of the amount of fluorescence in the microfluidic channel opposite the treated channel (Arbitrary Units, A.U.). Values are the mean ± S.E.M. of 5 independent experiments performed in triplicate. One-way ANOVA followed by Bonferroni’s showed no significance. Scale bar 30 μm. (**D**–**F**) On DIV 10, cortical and hippocampal primary cultures of neurons were transfected with siRNA for NR2A (blue), siRNA for NR2B (violet) or vehicle (black). Then, 24 h after, cells were treated with Aβ1-42 (500 nM) for 48 h and subsequently stimulated with vehicle or NMDA (15 µM) for 24 h. Neurite patterning was detected by immunocytochemistry using an anti-Nectin antibody (Abcam, 1/1000). Neurons were detected with the 3 anti-F-actin antibody fused to an Alexa 488 fluorophore (ThermoFisher, 1/400). Cell nuclei were stained with Hoechst (blue). Quantification was performed over segments of 15 μm. Each red dot represents a neurite formation. Values are the mean ± S.E.M. of 10 independent experiments performed in duplicates. One-way ANOVA followed by Bonferroni’s multiple comparison post hoc test was used for statistical analysis *** *p* < 0.001 versus vehicle treatment in control cells, ^&^
*p* < 0.05 versus NMDA treatment in control cells, ^#^
*p* < 0.05 versus Aβ1-42 treatment in control cells and ^$^
*p* < 0.05, ^$$^
*p* < 0.01 versus Aβ1-42 and NMDA treatment in control cells.

## Data Availability

The original contributions presented in the study are included in the article.

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
