# Peer review of "Dual Role of NMDAR Containing NR2A and NR2B Subunits in Alzheimer’s Disease"

_ijms, 2024, doi:10.3390/ijms25094757_

Round 1

Reviewer 1 Report

Comments and Suggestions for Authors

The manuscript presents a compelling investigation into the differential roles of NR2A and NR2B subunits in AD. The experiments are well-designed and the data suggest a neuroprotective role for NR2A and a detrimental role for NR2B. However, the authors can significantly strengthen their manuscript and ensure its clarity and impact for the scientific community by addressing some critical points.

Author Response

The manuscript by Raich et al. investigates the differential roles of NR2A and NR2B subunits of NMDA receptors in Alzheimer's disease (AD) using primary neuronal cultures. The findings suggest that NR2A activation protects neurons, while NR2B are more affected by Abeta oligomers, promoting cell death and cytotoxicity, indicating a different role between NR2A and NR2B in AD.

The manuscript is well organized and the experiments are for the majority well designed and described. However, to strengthen the manuscript for publication, I kindly ask the authors to address the following key points:

Major Concerns:

In the experiment on MAPK phosphorylation, the authors don't positively select transfected cells. They just indicated that they obtained a silencing efficiency greater than 80%. How can they be sure that the effects seen on MAPK phosphorylation are not due to a different number of silenced cells? Furthermore, the phosphorylation data are not normalized to the number of alive cells. How can the authors exclude that the decrease in phosphorylation observed in siRNA NR2A is due to the presence of NR2B alone and not to cell death? (Figure 3).

Thank you for the comment. It is true that would be perfect to have the possibility to just select the silenced cells. However, the amount of cells we require to do the experiment makes it impossible. However, we have repeated several times the quantification of NR2A and NR2B subunits expression in KD cells and considering a 20% expression respect 80% plus having all the negative and positive controls, we think we can distinguish between NR2A and NR2B expressions. About normalization, when we calculate the results, we always normalize our data to non-stimulated cells. Then, we avoid data from dead cells that do not signal.

In the experiments performed on primary cultures, for what reason did the authors decide to test Ki67, a marker of proliferation, on hippocampal neurons? Are these cortical and hippocampal neurons or cortical and hippocampal neural progenitors? Finally, despite recognizing the role of Caspase 3 in neuronal homeostasis, I believe that a single marker for such a complex phenomenon is extremely reductive. For this reason, I believe that it is necessary to consider at least one other marker to confirm the data obtained, such as the AMPK signaling pathway.

Thank you for the comment. Some authors propose that hippocampal cells can someway proliferate. Then we decided to test Ki67 marker. Moreover, we prepare primary cultures from fetuses, thus we could have some progenitor cells.

It would be nice to include new markers. However, to prepare primary cultures and test them we should need almost three months to generate fetuses, to do the dissection and then to process the samples several times. Moreover, the role of caspase 3 has been published and considered in many different manuscripts and the results seem consistent. Thus, we considered that caspase 3 results are enough to afford new data.

The data presented in Figure 4 are not described fully. How did the authors measure the AU? In the field? I believe that for this experiment, considering also the cell death caused by the Abeta treatment and therefore the possible difference in cell density in the different treatments, it would be better to indicate on the y-axis net intensity or fluorescence/cell.

Thank you for the comment. The fluorescence AU have been quantified considering different fields and dividing the amount of fluorescence per cell nuclei. The text has been modified introducing this consideration.

In the paragraph about the Abeta treatment that potentiates NMDAR containing NR2B, the authors speculate that the obtained results can be explained by the increase of NMDAR expression after the Abeta treatment (rows 273-274). Since the experiment is pretty straightforward, I suggest demonstrating this hypothesis. Moreover, this paragraph is missing a small conclusion to wrap it up.

Thank you for the comment. This paragraph has been rewritten to improve the comprehension of the presented results.

Unfortunately, the absence of a clear legend in the histogram graphs makes it difficult to read the data. I strongly suggest making the labels of the graphs clearer, in particular, those in Figures 3A, B, E, and F. Furthermore, in Figures 3G and H, the labels are missing in the WB panel.

And in all the graphs, the statistics are missing, even if reported in the figure legend.

Thank you for the comment. Figure 3 graph has been improved to clarify the results showed. Moreover, the statistics have been added in all graphs.

Minor Concerns:

-Row 223: specify in a better way the meaning of increase in NISSL staining, is the expression or the number of positive cells that increase?

Thank you for the comment. We have measured the staining and thus the protein expression.

-Row 225-226: what do the authors mean by "Correlation"? Does the NR2B subunit lead to a greater susceptibility/less resistance to Abeta treatment?

Thank you for the comment. It means that some Aß effects can be due to acting through NR2B subunit. Directly or indirectly. The text has been modified.

-Row 297-298: Add a citation.

Thank you for the comment. A citation has been included.

-Figure 5: the figure is cut on the right edge.

Thank you for the comment. The Figure has been improved.

-Figure 6D-E: explain the meaning of &, $, $$, and # in the figure legend.

Thank you for the comment. The Figure legend has been improved and better explained.

-Figure 6F is not cited in the text of the manuscript.

Thank you for the comment. Figure 6F has been cited.

By addressing these points, the authors can significantly strengthen their manuscript and ensure its clarity and impact for the scientific community.

Thank you once more for all the comments.

Reviewer 2 Report

Comments and Suggestions for Authors

NMDA receptors (NMDARs) play a pivotal role in Alzheimer’s Disease (AD) progression. They regulate calcium influx into neurons and exist as heterotetramers, with NR2A and NR2B subunits. NR2A activation promotes neuronal survival, while NR2B activation is linked to cell death. The study conducted by Raïch et al. investigated the signaling of NR2A and NR2B subunits in cortical and hippocampal neurons. They found that Aß, a key AD peptide, increases calcium release and MAPK phosphorylation via NR2B, leading to neuronal death. Conversely, NR2A promotes cell survival. In general, the study is performed linearly but several aspects needs to be supported statistically and should be extensively reviewed accordingly. My major comments are the following:

Result 1. NMDAR containing NR2A subunit shows stronger signalling than NMDAR containing NR2B subunit.

a)  The title is vague regarding the signaling promoted by the receptor.

b) NR2A-Rluc or NR2B-Rluc: Does Rluc stand for Rluciferase? Please provide more details. In addition, the description of the transfection method is missing, and the corresponding citation not only includes a wrong year but is also incongruous for the type of vector used, besides to referencing another article. The evaluation of receptor subunit expression through Rluc emission is not shown.

c) The data shown in Figure 1 lacks corresponding histograms for the representative kinetics and the related statistical analysis. Indicate the number of technical replicates, independent experiments, the applied statistical analysis for the experiments as in 1A, 1B, 1C.

Result 2. NMDA-induced toxicity in astroglia cells is due to NMDAR containing NR2B subunit.

a) Please provide immunoblotting data for siRNA transfection. The data reported in Figure 2A need to be supported by histograms and statistical analysis in order to demonstrate 1) the “significant” increase over basal levels in MAPK phosphorylation in control astrocytes or in the absence of NR2B as well as 2) the lower signal (only 150 % increase over basal levels) in NR2A KD astrocytes. Please, replace NR2A KO with NR2A KD.

b) The same statistical considerations apply to the data shown in Fig 2B obtained following pretreatment of astrocytes with LPS plus 200 U/mL IFN- γ for 48 h.

c) Cell viability test should be performed by a different approach (MTT, crystal violet, Alamar blue…)

d) For the statement “Lower but similar results were observed when astroglia was treated with NMDA 15 μM” no experimental data are showed.

Result 3. NMDA-induced cytotoxicity derives from stimulation of NMDAR containing NR2B subunit.

a) Please, explain the use of both cortical and hippocampal primary cultures of neurons. Does Alzheimer's disease preferentially affect hippocampal neurons compared to cortical neurons? Are specific markers known for the two populations? Please discuss this aspect.

b) The authors need to introduce the Aß1-42 peptide with references and rationale.

c) The data shown in Figure 3 requires a thorough statistical analysis without which the authors cannot draw conclusions. The Western blot data (Fig 3G, H) should be presented more rigorously, indicating the proteins detected and the molecular weights of the marker used, in addition to the gel-loaded samples. Did the revealed proteins (NMDAR subunits, ERK) belong to the same blot? Why the authors use ERK protein as normalizer? The legends of Fig. 3G,H seems to be wrong.

d) The viability test is not appropriate.

e) The data presented in Fig 4 needs to be presented more rigorously. The epifluorescence images need improvement in terms of resolution and magnification. The fluorescence analysis is not described, and the histograms lack statistical analysis, without which conclusions cannot be drawn. The authors claim to have conducted statistical analysis that they do not show.

Result 3. Aß treatment potentiates NMDAR containing NR2B subunit but not NMDAR containing NR2A subunit in primary neurons.

a) The statistical analysis for the experiments shown in Figure 5 is mandatory to discuss the obtained data.

Result 4. NMDAR activation does not affect Aß1-42 axonal transport.

The procedure used to estimate the axonal transport of Aß1-42 should be described in detail. Additionally, the displayed images are inadequate both in terms of magnification and style. Understanding of the phenomenon could be supported by lines/arrows or similar.

Line 303: Figure 5A-C is actually Figure 6A-C.

.

Result 5. NMDAR containing NR2A favours neurite formation in comparison to NMDAR containing NR2B.

a) The statistical analysis showed in Figure 6D-E seems to be confused as not properly described. The meaning of the symbols is not reported. It is not clear what the significance reported for the data regarding neurons transfected with siRNA for NR2A refers to. In addition, it is difficult to estimate the putative difference between the three control conditions in terms of neurite formation. Please, indicate in the histograms the populations of neurons analyzed. 

b) The author state “NR2A subunit shows a protective role in front of Aß1-42 toxicity. Contrary, NMDAR containing NR2B subunit is more sensible to Aß1-42 and NMDA cytotoxicity…”: It sounds inconsistent with the data emerging from the graphs.

Minor comments:

- Line 472: Please, control the citation (Navarro et al., 2015).

- Line 592: What microglia stand for?

- The authors should indicate the number of cells seeded per cm2 of surface area rather than per ml.

-Line 157: the authors state” Primary cultures of astroglial cells were transfected with siRNA for NR2A or siRNA for NR2B to knock down NR2A and NR2B expressions, respectively”. As the expression of both subunits were reduced but not completely abolished by RNA interference it is appropriate to refer to the astrocytes treated with siRNA as KD (knockdown) rather than KO (knockout).

Comments on the Quality of English Language

The English language require minor revision.

Author Response

NMDA receptors (NMDARs) play a pivotal role in Alzheimer’s Disease (AD) progression. They regulate calcium influx into neurons and exist as heterotetramers, with NR2A and NR2B subunits. NR2A activation promotes neuronal survival, while NR2B activation is linked to cell death. The study conducted by Raïch et al. investigated the signaling of NR2A and NR2B subunits in cortical and hippocampal neurons. They found that Aß, a key AD peptide, increases calcium release and MAPK phosphorylation via NR2B, leading to neuronal death. Conversely, NR2A promotes cell survival. In general, the study is performed linearly but several aspects needs to be supported statistically and should be extensively reviewed accordingly. My major comments are the following:

Result 1. NMDAR containing NR2A subunit shows stronger signalling than NMDAR containing NR2B subunit.

  1. a) The title is vague regarding the signaling promoted by the receptor.

Thank you for the comment. The title has been changed.

  1. b) NR2A-Rluc or NR2B-Rluc: Does Rluc stand for Rluciferase? Please provide more details. In addition, the description of the transfection method is missing, and the corresponding citation not only includes a wrong year but is also incongruous for the type of vector used, besides to referencing another article. The evaluation of receptor subunit expression through Rluc emission is not shown.

Thank you for the comment. Yes, Rluc means Rluciferase and the description has been included in the text. Also the description of the PEI transfection method has been included and a new citation. The values of Rluc bioluminescence emission have been included in the main text.

  1. c) The data shown in Figure 1 lacks corresponding histograms for the representative kinetics and the related statistical analysis. Indicate the number of technical replicates, independent experiments, the applied statistical analysis for the experiments as in 1A, 1B, 1C.

Thank you for the comment. The corresponding histograms for the representative kinetics and the related statistical analysis have been included. The number of technical replicates and independent experiments has been included in Figure 1A and 1B.

Result 2. NMDA-induced toxicity in astroglia cells is due to NMDAR containing NR2B subunit.

  1. a) Please provide immunoblotting data for siRNA transfection. The data reported in Figure 2A need to be supported by histograms and statistical analysis in order to demonstrate 1) the “significant” increase over basal levels in MAPK phosphorylation in control astrocytes or in the absence of NR2B as well as 2) the lower signal (only 150 % increase over basal levels) in NR2A KD astrocytes. Please, replace NR2A KO with NR2A KD.

Thank you for the comment. The immunoblotting data for siRNA transfection has been added and the histograms for data supported in Figure 2A and 2B have been included and statistical analysis have been done.

  1. b) The same statistical considerations apply to the data shown in Fig 2B obtained following pretreatment of astrocytes with LPS plus 200 U/mL IFN- γ for 48 h.

Thank you for the comment. The histogram for data supported in Figure 2B has been included and statistical analysis has been done.

  1. c) Cell viability test should be performed by a different approach (MTT, crystal violet, Alamar blue…)

Thank you for the comment. However, our extensive experience in the technique gives us the confidence with the obtained results. Moreover, then days do not give us the possibility to generate new primary astroglia to test the cultures. We would need almost two months.

  1. d) For the statement “Lower but similar results were observed when astroglia was treated with NMDA 15 μM” no experimental data are showed.

Thank you for the comment. These data is shown in Figure 2C.

Result 3. NMDA-induced cytotoxicity derives from stimulation of NMDAR containing NR2B subunit.

  1. a) Please, explain the use of both cortical and hippocampal primary cultures of neurons. Does Alzheimer's disease preferentially affect hippocampal neurons compared to cortical neurons? Are specific markers known for the two populations? Please discuss this aspect.

Thank you for the comment. This aspect has been discussed in the main text.

  1. b) The authors need to introduce the Aß1-42 peptide with references and rationale.

Thank you for the comment. The Aß1-42 peptide has been introduced in the main text.

  1. c) The data shown in Figure 3 requires a thorough statistical analysis without which the authors cannot draw conclusions. The Western blot data (Fig 3G, H) should be presented more rigorously, indicating the proteins detected and the molecular weights of the marker used, in addition to the gel-loaded samples. Did the revealed proteins (NMDAR subunits, ERK) belong to the same blot? Why the authors use ERK protein as normalizer? The legends of Fig. 3G,H seems to be wrong.

Thank you for the comment. Statistical analysis of Figure 3 has been included. The molecular weights have been included and the Figure legend has been modified according to the requirements. ERKs are proteins expressed in all cell types with similar levels and its antibody is pretty good. Then have been used as normalizer.

  1. d) The viability test is not appropriate.

Thank you for the comment. However, our extensive experience in the technique gives us the confidence with the obtained results. Moreover, then days do not give us the possibility to generate new primary astroglia to test the cultures. We would need almost two months.

  1. e) The data presented in Fig 4 needs to be presented more rigorously. The epifluorescence images need improvement in terms of resolution and magnification. The fluorescence analysis is not described, and the histograms lack statistical analysis, without which conclusions cannot be drawn. The authors claim to have conducted statistical analysis that they do not show.

Thank you for the comment. The images resolution has been improved. Also, we can give the original figures to the journal to increase the resolution. The statistics have been included to all histograms.

Result 3. Aß treatment potentiates NMDAR containing NR2B subunit but not NMDAR containing NR2A subunit in primary neurons.

  1. a) The statistical analysis for the experiments shown in Figure 5 is mandatory to discuss the obtained data.

Thank you for the comment. It is true that in dose-response curves it is difficult to develop statistical tests, however testing different concentrations of the compound instead of only one, like in bar graphs, shows a more solid and precise data.  

Result 4. NMDAR activation does not affect Aß1-42 axonal transport.

The procedure used to estimate the axonal transport of Aß1-42 should be described in detail. Additionally, the displayed images are inadequate both in terms of magnification and style. Understanding of the phenomenon could be supported by lines/arrows or similar.

Thank you for the comment. Additional information about microfluidic devices has been included in the text.

Line 303: Figure 5A-C is actually Figure 6A-C.

Thank you for the comment. The figure number has been modified.

Result 5. NMDAR containing NR2A favours neurite formation in comparison to NMDAR containing NR2B.

  1. a) The statistical analysis showed in Figure 6D-E seems to be confused as not properly described. The meaning of the symbols is not reported. It is not clear what the significance reported for the data regarding neurons transfected with siRNA for NR2A refers to. In addition, it is difficult to estimate the putative difference between the three control conditions in terms of neurite formation. Please, indicate in the histograms the populations of neurons analyzed.

Thank you for the comment. The Figure legend has been improved to include the different statistical tests. The populations of neurons have been included over histogram graphs.

  1. b) The author state “NR2A subunit shows a protective role in front of Aß1-42 toxicity. Contrary, NMDAR containing NR2B subunit is more sensible to Aß1-42 and NMDA cytotoxicity…”: It sounds inconsistent with the data emerging from the graphs.

Thank you for the comment. The text has been changed.. 

Minor comments:

- Line 472: Please, control the citation (Navarro et al., 2015).

Thank you for the comment. The citation has been removed.

- Line 592: What microglia stand for?

Thank you for the comment. There was a mistake between neurons and microglia.

- The authors should indicate the number of cells seeded per cm2 of surface area rather than per ml.

Thank you for the comment. The information has been modified.

-Line 157: the authors state” Primary cultures of astroglial cells were transfected with siRNA for NR2A or siRNA for NR2B to knock down NR2A and NR2B expressions, respectively”. As the expression of both subunits were reduced but not completely abolished by RNA interference it is appropriate to refer to the astrocytes treated with siRNA as KD (knockdown) rather than KO (knockout).

Thank you for the comment. KO has been modified for KD.

Reviewer 3 Report

Comments and Suggestions for Authors

This study builds on previous work from numerous labs on the role of the different NR2 subunits of the NMDA receptor in the response of cells to Abeta. For almost all of these studies, the authors use primary neurons separately derived from both cortex and hippocampus to address questions about the roles of NR2A and NR2B in the effects of Abeta on signaling, survival, axonal transport and neurite outgrowth. Overall, the experiments appear to be carefully done and the study makes a useful contribution to this field. However, I do have some concerns. First, since all of the experiments in the study rely on knocking down (or in Figure 1, overexpressing) NR2A and NR2B, the authors need to provide in supplementary data, figures showing the levels of knockdown or overexpression. Second, indicators of statistical significance are missing from most of the figures including Figures 1C, 2A,B, 3 (whole figure), 4 (whole figure) and 5 (whole figure). In addition, in Figure 6, the authors need to indicate what the different symbols refer to. Also, sometimes the authors refer to the knockdown itself and sometimes they refer to the consequences of the knockdown (ie effect of the expression of the receptor that is not knocked down). The manuscript would be easier to follow if the authors consistently used one or the other (either NR2A or NR2B ko or cells containing NR2A or NR2B). Some additional points that need to be addressed are listed below.

1. line 213: Please clarify if 42% and 65% represent survival or death.

2. Figure 3G,H is not mentioned in the text. Please include.

3. lines 315-318: The effects of NMDA and Abeta on NR2A knockout neurons seems much stronger in hippocampal as compared with cortical neurons. Thus, it is not clear why the authors don’t discuss this.

4. line 370: What did Abeta strongly potentiate?

5. line 403: “hat” should be “that”.

6. lines 406-409: It is not clear how the results from the neurite outgrowth studies shed light on a controversy regarding NMDARs and synapse loss. The authors need to clarify what the connection is between neurite formation and synapse loss.

7. line 434: Shouldn’t this be NR2A?

8. line 459: What is Ab3(pE)? What experiment was it used in?

9. lines 498 and 530: Information on the PEI method of transfection needs to be provided.

10. line 624: Why was a different transfection method used for these studies? Was the level of knockdown similar to that found with the PEI method?   

Comments on the Quality of English Language

The English needs general editing as word usage is quite awkward throughout.

Author Response

This study builds on previous work from numerous labs on the role of the different NR2 subunits of the NMDA receptor in the response of cells to Abeta. For almost all of these studies, the authors use primary neurons separately derived from both cortex and hippocampus to address questions about the roles of NR2A and NR2B in the effects of Abeta on signaling, survival, axonal transport and neurite outgrowth. Overall, the experiments appear to be carefully done and the study makes a useful contribution to this field. However, I do have some concerns.

First, since all of the experiments in the study rely on knocking down (or in Figure 1, overexpressing) NR2A and NR2B, the authors need to provide in supplementary data, figures showing the levels of knockdown or overexpression.

Thank you for the comment. Levels of receptor expression in transfected HEK-293T cells experiments have been added to the text. Western blot images and further quantification of NR2A and NR2B KD in neuronal primary cultures have been included in Figure 3.

 Second, indicators of statistical significance are missing from most of the figures including Figures 1C, 2A,B, 3 (whole figure), 4 (whole figure) and 5 (whole figure). In addition, in Figure 6, the authors need to indicate what the different symbols refer to.

Thank you for the comment. All missing information has been included in the figures and in the text.

Also, sometimes the authors refer to the knockdown itself and sometimes they refer to the consequences of the knockdown (ie effect of the expression of the receptor that is not knocked down). The manuscript would be easier to follow if the authors consistently used one or the other (either NR2A or NR2B ko or cells containing NR2A or NR2B). Some additional points that need to be addressed are listed below.

Thank you for the comment. The word KO (knockout) has been exchanged to KD (knockdown) to improve the manuscript comprehension.

  1. line 213: Please clarify if 42% and 65% represent survival or death.

 Thank you for the comment. The information has been clarified.

  1. Figure 3G,H is not mentioned in the text. Please include.

  Thank you for the comment. The Figure 3G,H has been mentioned.

  1. lines 315-318: The effects of NMDA and Abeta on NR2A knockout neurons seems much stronger in hippocampal as compared with cortical neurons. Thus, it is not clear why the authors don’t discuss this.

Thank you for the comment. The sentence has been explained.

  1. line 370: What did Abeta strongly potentiate?

 Thank you for the comment. Aß potentiates NMDAR containing NR2B signalling. The text has been modified.

  1. line 403: “hat” should be “that”.

  Thank you for the comment. The mistake has been corrected.

  1. lines 406-409: It is not clear how the results from the neurite outgrowth studies shed light on a controversy regarding NMDARs and synapse loss. The authors need to clarify what the connection is between neurite formation and synapse loss.

  Thank you for the comment. The sentence has been modified. Is true that we do not shed light, we just afford new data to the role of NMDAR subunits in AD.

  1. line 434: Shouldn’t this be NR2A?

   Thank you for the comment. That’s true. The word has been exchanged.

  1. line 459: What is Ab3(pE)? What experiment was it used in?

 Thank you for the comment. There was an extra paragraph that has been removed.

  1. lines 498 and 530: Information on the PEI method of transfection needs to be provided.

 Thank you for the comment. This information has been included. 

  1. line 624: Why was a different transfection method used for these studies? Was the level of knockdown similar to that found with the PEI method?

 Thank you for the comment. siRNA expression is a bit more efficient with lipofectamine that with PEI. Then, we employ this method to Knockdown, and PEI to overexpress receptors. The text has been modified and lipofectamine protocol has been included.

Round 2

Reviewer 1 Report

Comments and Suggestions for Authors

With the improvements made by the authors I consider this manuscript ready for publication.